# True Donor Cell Leukemia after Allogeneic Hematopoietic Stem Cell Transplantation: Diagnostic and Therapeutic Considerations—Brief Report

Michèle Hoffmann [1], Yara Banz [2], Jörg Halter [3], Jacqueline Schoumans [4], Joëlle Tchinda [5,6], Ulrike Bacher [6,†] and Thomas Pabst [1,*,†]

1    Department of Medical Oncology, Inselspital Bern, Bern University Hospital, University of Bern, 3012 Bern, Switzerland; michele.hoffmann@insel.ch
2    Institute of Tissue Medicine and Pathology, University of Bern, 3012 Bern, Switzerland; yara.banz@unibe.ch
3    Division of Hematology, University Hospital Basel, 4031 Basel, Switzerland; joerg.halter@usb.ch
4    Oncogenomic Laboratory, Service and Central Laboratory of Hematology, University Hospital Lausanne (CHUV), 1005 Lausanne, Switzerland; jacqueline.schoumans@chuv.ch
5    Laboratory for Oncology, University Children's Hospital Zürich, 8032 Zürich, Switzerland; joelle.tchinda@kispi.uzh.ch
6    Department of Hematology and Central Laboratory, Inselspital Bern, Bern University Hospital, University of Bern, 3012 Bern, Switzerland; veraulrike.bacher@insel.ch
\*    Correspondence: thomas.pabst@insel.ch; Tel.: +41-31-632-0378; Fax: +41-31-632-3410
†    These authors contributed equally to this work and are both considered last authors.

**Abstract:** Donor cell leukemia (DCL) is a rare complication after allogeneic hematopoietic stem cell transplantation (HSCT) accounting for 0.1% of relapses and presenting as secondary leukemia of donor origin. Distinct in phenotype and cytogenetics from the original leukemia, DCL's clinical challenge lies in its late onset. Its origin is affected by donor cell anomalies, transplant environment, and additional mutations. A 43-year-old woman, treated for early stage triple-negative breast cancer, developed mixed-phenotype acute leukemia (MPAL), 12 years later. Following induction chemotherapy, myeloablative conditioning, and allo-HSCT from her fully HLA-matched brother, she exhibited multiple cutaneous relapses of the original leukemia, subsequently evolving into DCL of the bone marrow. Cytogenetic analysis revealed a complex male karyotype in 20 out of 21 metaphases, however, still showing the MPAL phenotype. DCL diagnosis was confirmed by 90.5% XY in FISH analysis and the male karyotype. Declining further intensive chemotherapy including a second allo-HSCT, she was subsequently treated with repeated radiotherapy, palliative systemic therapies, and finally venetoclax and navitoclax but died seven months post-DCL diagnosis. This case underlines DCL's complexity, characterized by unique genetics, further complicating diagnosis. It highlights the need for advanced diagnostic techniques for DCL identification and underscores the urgency for early detection and better prevention and treatment strategies.

**Keywords:** donor cell leukemia; allogeneic hematopoietic stem cell transplantation; mixed-phenotype acute leukemia; chimerism; leukemogenesis

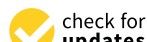



## 1. Introduction

Donor cell leukemia (DCL) is a rare and specific potential consequence following allogeneic hematopoietic stem cell transplantation. The hallmark is the development of a secondary leukemia of donor origin. This condition accounts for as few as 0.1% of relapses after allogenic stem cell transplantation in acute and chronic leukemias, with a notable increase in reported cases over the recent 50 years [1]. DCL typically harbors distinct cytogenetic and phenotypic profiles compared to the initial disease in a given patient, contributing to eventual disease heterogeneity between DCL and the initial leukemia subtype. Whereas acute myeloid leukemia (AML) represents about 50% of all reported DCL cases, DCL can

also be observed in acute lymphoblastic leukemia (ALL) and myelodysplastic syndrome (MDS), with a typically prolonged latency period to diagnosis of a median of 26 months [2]. This late manifestation implies significant changes in disease characteristics. Whilst the phenomenon of DCL is rare, the diagnostic and therapeutic challenges justify awareness for this unique condition.

The development of donor cell leukemia (DCL) involves the interplay between pre-existing preleukemic clones in the donor hematopoiesis and additional transplant-related factors. This process is affected by the transplant environment, particularly when stem cells are positioned in a permissive microenvironment, with subsequent bone marrow injury from conditioning chemotherapy and an immature or suppressed immune system. Moreover, the variable latency and incomplete penetrance of hematologic neoplasms suggest that additional somatic mutational events are required for the initiation and development of DCL.

The clinical course of DCL is mostly dismal, with a 47% mortality after a follow-up of 8.5 months in one report, highlighting the need for a better understanding of the causative factors of DCL, for more effective treatment options, but also for an earlier detection and eventually preventive actions [2,3]. Here, we report a patient with DCL who refused a second allogeneic stem cell transplantation and favored a personalized treatment approach based on the results of pharmacoscopy-based drug sensitivity screening.

## 2. Case Report

In February 2017, a 43-year-old Caucasian woman was referred to our center with fatigue and vaginal bleedings despite endometrial ablation 3 months ago. The medical history of the patient was notable by an early stage triple-negative breast cancer (pT2 pN0(0/4) cM0 R0 G2) diagnosed 12 years earlier. She underwent primary surgical resection, followed by adjuvant chemotherapy with four cycles of farmorubicin and cyclophosphamide, completed with additional four cycles of paclitaxel. Following chemotherapy, she underwent radiotherapy with a total dose of 60 Gy.

The initial diagnostic work-up identified thrombocytopenia with a platelet count of $30 \times 10^9$ cells/L, anemia with hemoglobin of 9.7 g/dL, and a white cell count of $9.05 \times 10^9$ cells/L with 53% blast cells in peripheral blood. The absolute neutrophil count was $2.94 \times 10^9$ cells/L (normal range: $1.6 \sim 7.4 \times 10^9$ cells/L). Coagulation parameters and biochemistry were normal except for elevated serum lactate dehydrogenase (LDH 679 U/L; normal range < 480 U/L). Bone marrow biopsy showed a predominant blast infiltration of 90–100%. Immunophenotyping characterized the blasts as mixed-phenotype acute leukemia (MPAL), T/myeloid, NOS according to the current WHO classification [4] with CD45(+), TdT+, cCD3+, sCD3+, CD5+, CD7+, CD13+, CD117+, MPO+, CD38+, HLA-DR-, CD1a partially weak positive, and negative B-cell markers (Figure 1). Cytogenetic analysis showed a female complex karyotype with trisomy 8 and monosomies 11, 13, 14, 16 (47,XX,add(5)(q31),add(7)(p10),+8,−11,−13,−14,−16,+4mar[3]/46,XX[17]) (Figure 2A). Next, generation sequencing (NGS) analysis of the bone marrow aspirate identified two mutations in the *NOTCH1* gene, both with a variant allele frequency (VAF) of 43%: c.7413_7416delinsCTTTACCTTCTC, p.(Leu2472Phefs*8) and c.5126T > C, p.(Leu1709Pro) (Figure 1). *BCR-ABL1* was negative. Two distinct minimal residual disease (MRD) markers were discerned: a T-cell receptor delta (V1J/JD1) rearrangement and a T-cell receptor beta (Db1/Jb1.2) rearrangement, respectively. Cerebrospinal fluid (CSF) analysis excluded leukemic involvement of the central nervous system (CNS). Considering the history of breast tumor treated with chemotherapy and radiotherapy, therapy-associated leukemia was assumed.

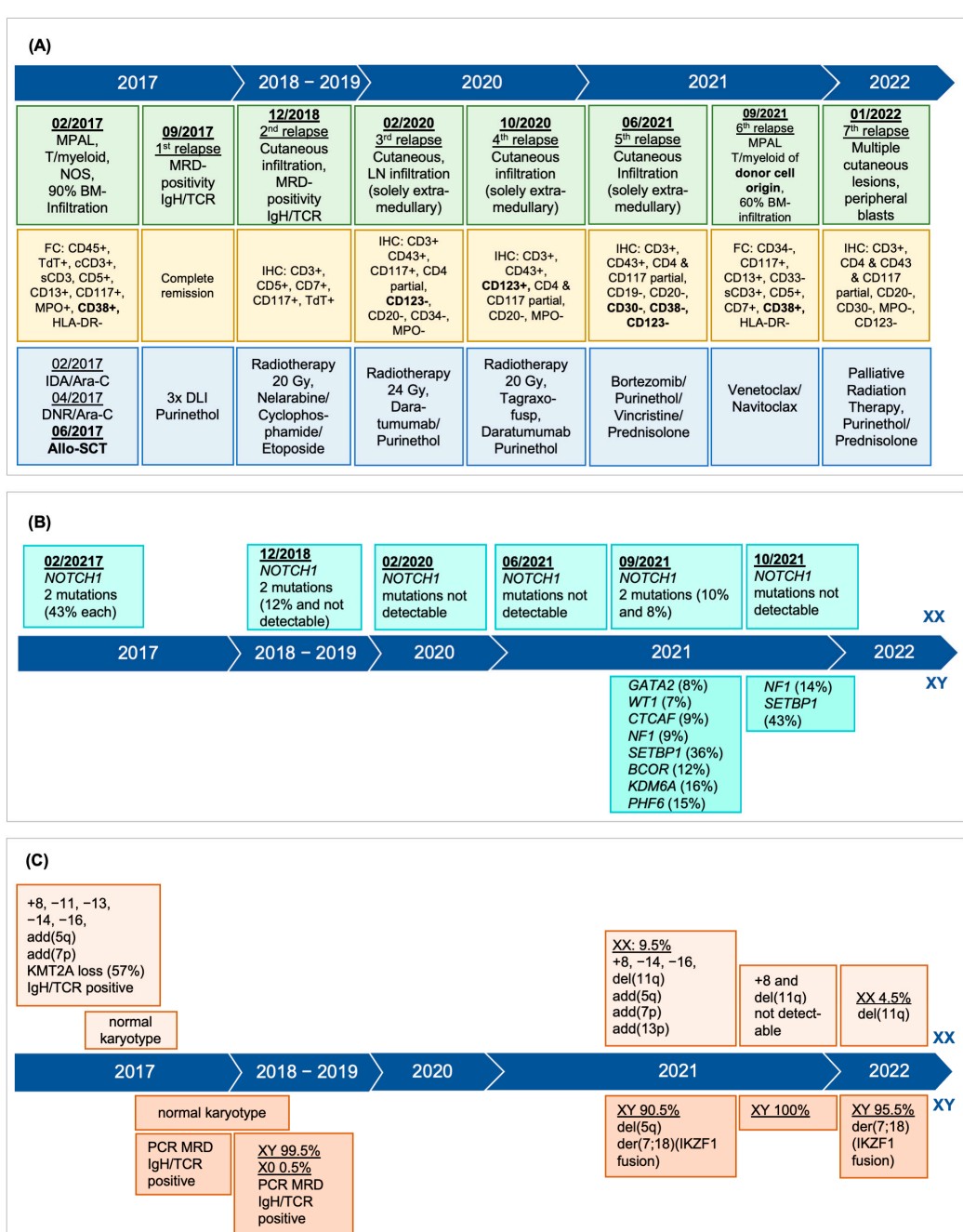

**Figure 1.** Timeline composite figure indicating the clinical manifestations, results of flowcytometry (FC) and immunohistochemistry (IHC) and treatment (**A**), molecular diagnostics (**B**), and cytogenetics and FISH analysis (**C**).

The patient received intensive induction chemotherapy with one cycle of cytarabine/idarubicin followed by one cycle of cytarabine/daunorubicin. The treatment was well tolerated, and the patient achieved a complete morphological and cytogenetic remission after cycle 1, which was confirmed after cycle 2. Considering the adverse risk category according to the ELN risk classification, the patient underwent subsequent allogeneic hematopoietic stem cell transplantation (allo-SCT) from her HLA-identical 61-year-old brother in June 2017. The myeloablative conditioning regimen included cyclophosphamide and busulfan. Graft-versus-host disease (GvHD) prophylaxis consisted of a combination of ATG, cyclosporine A (CyA), and methotrexate. Following hematopoietic engraftment, flow-cytometric and molecular MRD-negative complete remission with a 100% CD34+ donor

chimerism was documented in whole blood and bone marrow (CD3+ and CD66+ cells) one month after transplant. Molecular MRD analysis of the immunoglobulin heavy-chain (IgH) and T-cell receptor (TCR) gene rearrangements was negative.

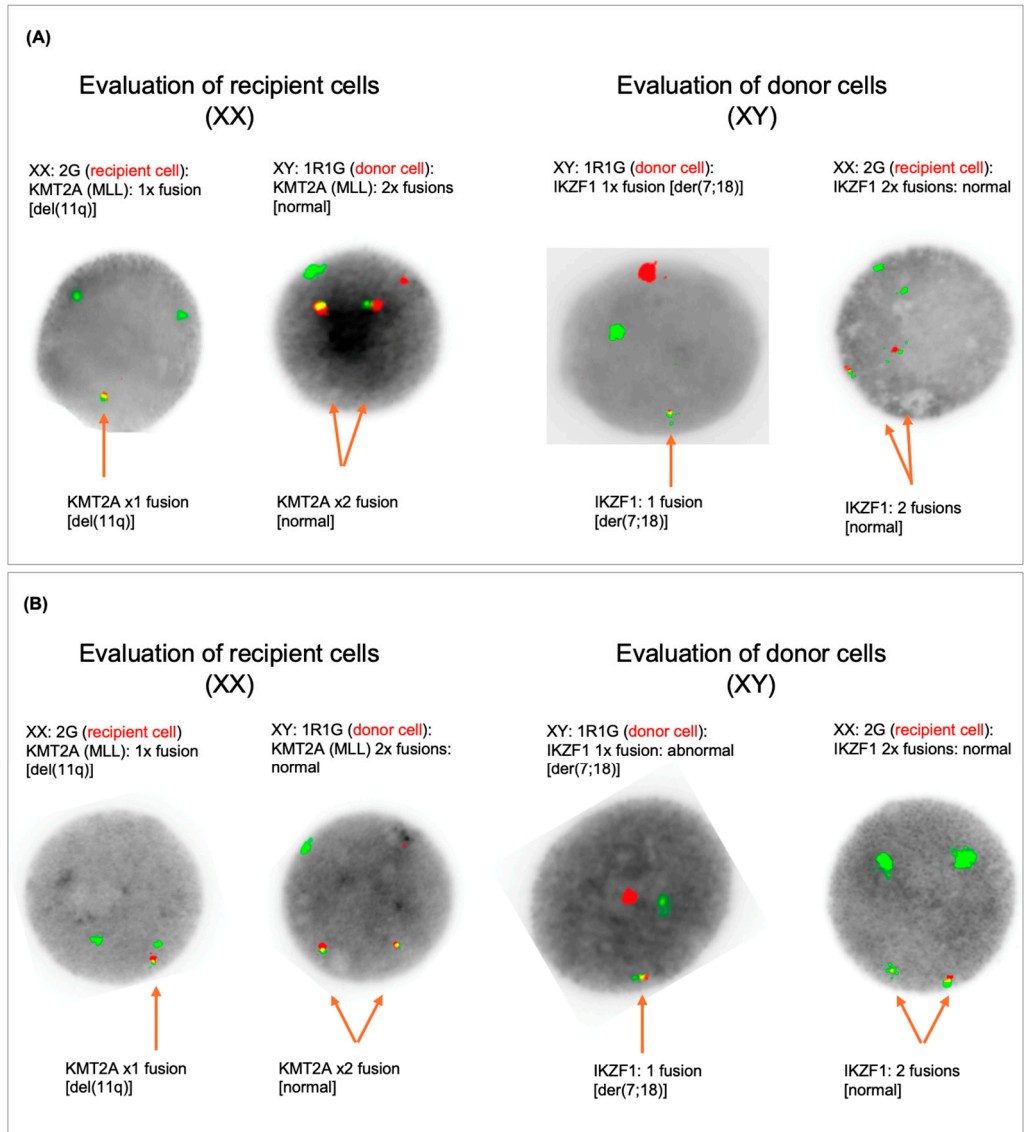

**Figure 2.** FISH analysis from the bone marrow at the time of diagnosis of the DCL: All male (donor) metaphases showed a whole-arm translocation between chromosome 7 and 18 leading to a loss of 7p and 18p. The presence of a complex female karyotype with the aberrations (i.e., monosomy 11) seen at presentation is consistent with relapse of the BAL. The aberrant complex male karyotype and the whole-arm translocation between chromosome 7 and 18 leading to a loss of 7p and 18p indicate a donor cell leukemia (**A**). FISH analysis from peripheral blood (7th relapse, DCL): FISH analysis was performed on uncultured peripheral blood cells with the XL MLL plus break-apart probe and centromere for chromosomes X and Y (MetaSystems GmbH) and IKZF1 Break Probe (Cytocell). In 200 nuclei analyzed for each region tested, a signal pattern was detected of female cells with a deletion of 11q in 4.5% and male cells with deletion of 7p in 86.5% (**B**).

Three months after allo-SCT, bone marrow biopsy suggested early molecular relapse with MRD positivity (PCR analysis of IgH and TCR rearrangement), but ongoing morphologic complete remission. Accordingly, immunosuppression with cyclosporine A was stopped and treatment with purinethol® (mercaptopurine, Aspen Pharma, Baar, Switzerland) was initiated, given the initial interpretation of T-ALL. In addition, three donor

lymphocyte infusions from the HLA-identical brother (initial donor) induced, again, an MRD-negative complete remission in April 2018. Remarkably, we observed no evidence of graft-versus-host disease (GvHD) at any stage.

From December 2018 until July 2021, repeated radiotherapy interventions and various systemic therapies were performed to control multiple, localized, histologically confirmed, cutaneous recurrences of the initial leukemia. An extensive laboratory work-up as well as bone marrow analyses performed at each relapse again showed a molecular relapse with MRD positivity (PCR analysis of IgH/TCR rearrangement) in 12/2018 with a morphological and flowcytometric complete remission (Figure 1) and an ongoing 100% CD34+ cell donor chimerism.

In September 2021, 4 years and 3 months after allo-SCT, the patient presented with aggravated thrombocytopenia of $46 \times 10^9$/L and with 3.5% blast cells in peripheral blood. In the bone marrow, a 60% infiltration by blast cells, immunohistochemically positive for CD3, CD117+, CD13+, sCD3+, CD5+, CD7+, CD38+ and negative for CD34, CD 33, HLA-DR, was identified. The leukemia-associated immunophenotype (LAIP) was consistent with a relapse of MPAL T/myeloid. *NOTCH1, GATA2, WT1, CTCF, NF1 STBP1, BCOR, KDM6A,* and *PHF6* genes were mutated via NGS analysis (Figure 1). We identified two specific MRD markers, based on the genomic DNA sequence of a TCR-delta (V1J/JD1) and a TCR-beta (Db1/Jb1.2) rearrangement, respectively. These DNA markers demonstrated a quantifiable range in PCR analysis of $1.0 \times 10^{-4}$ (V1J/JD1) and $5.0 \times 10^{-4}$ (Db1/Jb1.2), with a sensitivity of $1.0 \times 10^{-4}$ (V1J/JD1) and $1.0 \times 10^{-5}$ (Db1/Jb1.2). Karyotype was as follows: (46,XX,add(5)(q35),add(7)(p13),+8,del(11)(q14-22),add(13)(p11.1),−14,−16,+mar[1]// 45,XY,del(5)(q12q14-21),der(7;18)(q10;q10)[20]). However, and intriguingly, the cytogenetic analysis suggested donor cell origin of this mixed-phenotype leukemia due to an aberrant complex male karyotype in the vast majority of the metaphases (20 out of 21) (Figures 1 and 2A). In addition, the simultaneous presence of a complex female karyotype with monosomy 11 in 1 out of 21 metaphases as seen at initial presentation as well as the immunophenotype with myeloid and T-cell features was consistent with relapse of the initial MPAL. All donor metaphases (male karyotype) showed interstitial deletion of 5q and a whole-arm translocation between chromosome 7 and 18 leading to a loss of 7p and 18p. The karyotype together with the fluorescent in situ hybridization (XY: 90.5%), confirmed the donor cell origin (patient's brother) of the leukemia in a high proportion of leukemia cells. Unfortunately, we were unable to re-evaluate the donor (brother of the patient) at the time of diagnosis of the donor cell leukemia since he died in an accident June 2018 with no known hematologic pathology. Family history was unremarkable regarding malignant disorders.

Based on drug sensitivity screening data performed with pharmacoscopy using real-time patient-derived leukemic samples to generate treatment recommendations [5], therapy was switched to venetoclax in combination with a low dose of navitoclax in September 2021. A bone marrow examination on day 27 of the first cycle showed an outstanding therapeutic response, achieving a morphologic complete remission with immunophenotypic MRD negativity, while the FISH analysis confirmed the presence of male donor cells (100%).

Four months later, January 2022, a seventh relapse was diagnosed with rapid progression and 20% peripheral blasts. Immunophenotyping and FISH analysis confirmed predominant donor cell leukemia, type MPAL. FISH analysis confirmed that the recipient cells showed the deletion of 11q and that the donor cells had a male karyotype with deletion of 7p due to der(7;18) (Figure 2B). Notably, there was an extensive cutaneous extramedullary manifestation of the right lower leg, for which palliative radiotherapy was started since the patient declined further systemic treatment. Additional new lesions developed in the skin and gingiva. Ultimately, the patient died on April 2022, a total of seven months after the diagnosis of donor cell leukemia and 5 years after the initial diagnosis of MPAL.

### 3. Discussion and Conclusions

Since the first description of DCL in 1971 [6], more than 100 cases have been reported, predominantly among bone marrow transplant recipients. DCL has both been observed in allogeneic peripheral blood stem cell transplantations and in cord blood transplants. The umbilical cord seems to present a potentially higher risk for DCL compared to other stem cell sources [7]. The European Society for Blood and Marrow Transplantation (EBMT) estimates the incidence of DCL in allo-HSCT recipients to be approximately 0.1% [1]. Historically, the diagnosis of DCL was challenging, relying on morphological differences or karyotype analysis, which were not always available for distinguishing between donor and recipient cells. Recent advances in molecular short tandem repeat (STR) analysis and sensitive flow cytometry complemented by chimerism studies using CD34+ cells in peripheral blood have improved the accuracy in determining the donor origin of leukemic cells, contributing to a more precise incidence assessment of DCL [8]. Case studies and a review of the literature indicate a rise in the incidence of DCL since 2004, most likely due to improvements in detection methods, including molecular analysis and cytogenetic advancements [9].

In the patient presented in this report, the sex mismatch facilitated the identification of the donor cell origin, as evidenced by karyotype and FISH. In sex-matched allo-transplant scenarios, the diagnosis of DCL is more demanding, and cytogenetic analysis and chimerism are less effective. Precise molecular diagnosis is crucial for identifying the donor origin of leukemia.

Acute myeloid leukemia (AML) and acute lymphocytic leukemia (ALL) are the most common forms of DCL. The onset latency of DCL may vary from a few months to more than ten years, implying a multifactorial developmental process involving hematopoietic stem cells (HSCs), the microenvironment, and various oncogenic factors [1,2,9–11]. In the patient reported here, time to donor cell leukemia was 4 years and 3 months.

Mutations in stem or progenitor cells, whether present in donor cells or newly emerging post-transplantation due to factors like residual bone marrow injury from therapy, reduced immune monitoring, and replicative stress can disrupt regular cell functions. This disruption can lead to a clonal expansion and trigger the onset of leukemia. Interestingly, the age of the donor seems not to affect the onset for the development of donor cell-derived hematologic neoplasms (DCHN) [2].

The detection of mutations typically associated with pre-leukemic phases resonates with the current evidence indicating a high frequency of such mutations in individuals over 60 years, heralding an elevated risk for hematologic malignancies. Multiple studies have indicated the presence of clonal hematopoiesis, marked by somatic mutations, in approximately 10% of older individuals. It became clear that these somatic mutations, predominantly involving genes like *DNMT3A*, *TET2*, and *ASXL1*, were associated with an increased risk of hematologic cancers, a higher overall mortality, and an elevated risk of cardiometabolic diseases [12]. Given this trend, screening older donors for such mutations may appear reasonable as they are at a greater risk of carrying undetected malignant cell clones, but data for such an approach are largely missing and most of these donors do ultimately not evolve into clinically overt hematologic malignancies. In conclusion, screening donors for CHIP and its emerging consequences remains controversial [12–15]. A thorough assessment of the donor's family medical history is important, especially concerning leukemia and other hematological malignancies. This can be helpful to identify potential genetic predispositions of the donor. Donor cell leukemia is an exceedingly rare phenomenon, posing significant challenges in establishing preventive strategies for this condition.

In their systematic review, Suàrez et al. examined the health conditions of donors at the point of diagnosis for donor cell hematologic neoplasms. Out of 73 cases, 85% of the donors remained healthy, whereas 12% progressed towards hematologic malignancies and 3% were diagnosed with non-hematologic cancers. The review also highlighted that, in cases where comprehensive molecular analyses were performed, certain mutations present

in leukemic cells at the time of DCHN were identified in the donors. Remarkably, only two of these donors subsequently developed a hematologic malignancy. Furthermore, in an additional seven cases where donors developed hematologic malignancies but without the benefit of molecular studies, the type of malignancy corresponded with that of the recipients [2]. Finding specific mutations in leukemic cells at DCHN diagnosis that were also present in donors indicates a possible inherited or acquired genetic predisposition to the development of hematologic neoplasms.

Based on a systematic review of DCHN after hematopoietic stem cell transplantation, the most recurrent cytogenetic abnormalities in DCHN involved chromosome 7 deletions, anomalies of chromosome 5, or complex karyotypes, which are commonly seen in therapy-related myeloid neoplasms [2].

In our case, cytogenetic analysis showed two genetic alteration patterns and a complex karyotype. The first contained a complex karyotype with trisomy 8, monosomies 14 and 16, and deletion 11q corresponding to the patient/recipient hematopoiesis at the time of the initial acute leukemia. The second clone was characterized by an interstitial deletion of 5q and a whole-arm translocation between chromosome 7 and 18 leading to a loss of 7p and 18p corresponding to the donor/male leukemic cells. This may allow the hypothesis that a deficiency in the immune system of the recipient is responsible for the alterations in her own hematopoiesis and later in the donor hematopoiesis following allogeneic stem cell transplantation. Alternatively, one may postulate that the brother's hematopoiesis also harbored clonal changes at a very low level that increased after allogeneic transplantation in the recipient under immunosuppression, while no obvious hematologic disease was known in the donor until his (unrelated) death at a young age. This early death unfortunately prohibited donor re-evaluation. A cytogenetic analysis from bone marrow performed one and three months after allogeneic stem cell transplantation showed a normal (male) karyotype. To which extent the patient's post-transplant treatment have contributed to this development remains a matter of speculation.

Treatment approaches for donor cell leukemia (DCL) remain poorly studied, likely due to the condition's scarcity and diverse clinical presentation. Therapeutic options may range from chemotherapy and targeted therapies to additional allogeneic stem cell transplants from a new donor or from the same geno-identical donor. While the prognosis for DCL is usually poor, emerging developments in genomic profiling and targeted treatment approaches may provide promising opportunities for more effective interventions in the future.

The standard therapy for donor cell leukemia to date typically involves reinduction chemotherapy and, again, allogeneic transplantation if curation is intended, employing regimens specific to acute myeloid leukemia (AML) or acute lymphoblastic leukemia (ALL). Sustained remissions are achievable, especially when consolidated with a second HSCT, with a mean overall survival of 32.8 months [9].

The DARTT-1 Trial, a prospective, non-randomized, single-center observational study showed that performing pharmacoscopy in patients with AML at relapse who had exhausted all registered treatment options appears beneficial with a median overall survival of 18 weeks [5]. In our case, the use of venetoclax, a selective BCL2 inhibitor in combination with a low dose of navitoclax, a BCL-XL/BCL inhibitor, following an innovative precision medicine approach ("pharmacoscopy"), complemented by radiotherapy of the cutaneous leukemia manifestations, represented a personalized (palliative) treatment approach since our patient refused further intensive systemic chemotherapy and second allogeneic stem cell transplantation, respectively. This approach facilitated a targeted treatment strategy based on the unique characteristics of the patient's leukemia. Unfortunately, the patient died seven months following the diagnosis of DCL, which is consistent with a median survival of 5.5 months as reported in the literature [9].

Particularly noteworthy in our case is the preservation of the MPAL despite the predominant presence of a new "donor-derived" male karyotype in 20 out of 21 metaphases. This observation is extraordinary as the genetic constitution of leukemia cells is typically

distinctly attributed to either the original recipient or the donor. The case exhibits an unusual amalgamation of both genetic profiles, suggesting that the recipient's leukemic cells were influenced by the donor's cells, leading to a change in the karyotype while preserving the MPAL phenotype. This phenomenon illustrates the complexity of interactions between donor and recipient cells and underscores the need for precise analysis of genetic and phenotypic changes in cases of DCL.

**Author Contributions:** Conceptualization, T.P., M.H. and U.B.; methodology, T.P. and U.B.; validation, T.P., M.H. and U.B.; formal analysis, M.H.; investigation, M.H., J.T., Y.B., J.H. and J.S.; resources, Y.B., J.T., J.S. and U.B.; data curation, T.P.; writing—original draft preparation, M.H.; writing—review and editing, U.B., M.H. and T.P.; supervision, T.P.; project administration, T.P. and U.B. All authors have read and agreed to the published version of the manuscript.

**Funding:** This research received no external funding.

**Institutional Review Board Statement:** Not applicable.

**Informed Consent Statement:** Informed consent was obtained from the patient to publish this paper.

**Data Availability Statement:** The data presented in this study are available upon request from the corresponding author.

**Acknowledgments:** The authors wish to thank the patient and its family for the approval to report these data. Also, the authors thank all the physicians and nurses involved in the care of this patient.

**Conflicts of Interest:** All authors declared no financial relationships with any organizations that might have an interest in the submitted work in the recent three years and no other relationships or activities that could appear to have influenced this work.

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
