# Peer review of "True Donor Cell Leukemia after Allogeneic Hematopoietic Stem Cell Transplantation: Diagnostic and Therapeutic Considerations—Brief Report"

_curroncol, doi:10.3390/curroncol31040153_

Round 1

Reviewer 1 Report

Comments and Suggestions for Authors

Hoffmann M et al. report on a patient who developed donor cell leukaemia, a rare but important complication after allogeneic haematopoietic stem cell transplantation.

They studied immunophenotypic, cytogenetic and genetic alterations of the leukaemic cells in detail in the patient.

Their efforts and interpretation were excellent and well documented.

I think this case report should be published in Current Oncology as soon as possible.

Minor comments

1, What about family history of cancer?

2, Were clonal TCR and IGH rearrangements detected in the initial leukaemia presented in 02/2017? 

Author Response

Reviewer 1:

Hoffmann M et al. report on a patient who developed donor cell leukaemia, a rare but important complication after allogeneic haematopoietic stem cell transplantation.

They studied immunophenotypic, cytogenetic and genetic alterations of the leukaemic cells in detail in the patient.

Their efforts and interpretation were excellent and well documented.

I think this case report should be published in Current Oncology as soon as possible.

Minor comments:

  1. What about family history of cancer?

Authors: The reviewer addressed a very important point. The patient had no family history of cancer. We added this relevant detail (lines 148-149).

  1. Were clonal TCR and IGH rearrangements detected in the initial leukaemia presented in 02/2017? 

Authors: We agree with the reviewer that this information is very important and added the details of TCR rearrangements used for MRD detection to lines 89-92.

Reviewer 2 Report

Comments and Suggestions for Authors Hoffman and colleagues present a very interesting case report of a breast cancer patient who developed MPAL, went on to receive a MSD-HSCT, and then develop donor derived leukemia, but also had MRD persistence of her original MPAL leukemia. They present compelling evidence that the second leukemia was indeed donor derived and discuss a review of the known literature.    Overall, the case report is easy to read and it adds value to the existing literature.  Because of the donor-recipient sex-mismatch, the DCL was easy to identify and did not require STR analysis.  Unfortunately, the donor was not available for follow-up studies for reasons mentioned in the report.    A few minor comments/questions:  1.  Line 101. The report states 100% CD34+ donor chimerism - can you explain if this was done in whole blood or bone marrow and was lineage-specific (T or Myeloid) chimerisms done? Line 117 points out that donor chimerism is shown in Figure 1, but the only thing I see in Panel C of Figure 1 is the Cytogenetics and FISH analysis of the DCL.     2. Line 108: Purinethol. Please indicate beside in parentheses "(mercaptopurine)" for North American readers.    2.  Line 63 alludes to preventive actions for DCL.  What is/are available and maybe this can be expanded in the Discussion.    3. Please indicate the age of the donor at the time of HSCT.

Author Response

Reviewer 2:

Hoffman and colleagues present a very interesting case report of a breast cancer patient who developed MPAL, went on to receive a MSD-HSCT, and then develop donor derived leukemia, but also had MRD persistence of her original MPAL leukemia. They present compelling evidence that the second leukemia was indeed donor derived and discuss a review of the known literature. Overall, the case report is easy to read and it adds value to the existing literature. Because of the donor-recipient sex-mismatch, the DCL was easy to identify and did not require STR analysis. Unfortunately, the donor was not available for follow-up studies for reasons mentioned in the report.   

A few minor comments/questions: 

  1. Line 101. The report states 100% CD34+ donor chimerism - can you explain if this was done in whole blood or bone marrow and was lineage-specific (T or Myeloid) chimerisms done?

Authors: Chimerism was done in whole blood and bone marrow (CD3+ and CD66+ cells). This was now mentioned in detail.

  1. Line 117 points out that donor chimerism is shown in Figure 1, but the only thing I see in Panel C of Figure 1 is the Cytogenetics and FISH analysis of the DCL.    

Authors: Figure 1, referenced in line 118, illustrates the repeated radiotherapy interventions, various systemic treatments, and the comprehensive laboratory work-up and bone marrow analysis conducted at each relapse. However, chimerism is not shown in Figure 1. We agree with the reviewer's observation that this omission may cause confusion and have accordingly mentioned Figure 1 earlier in the sentence, alongside the description of treatments and analyses.

  1. Line 108: Purinethol. Please indicate beside in parentheses "(mercaptopurine)" for North American readers.   

Authors: We followed this advice (line 108).

  1. Line 63 alludes to preventive actions for DCL. What is/are available and maybe this can be expanded in the Discussion.   

Authors: We thank the reviewer for this good comment. We have elaborated this important topic in the discussion section line 208-219. Various studies have observed clonal hematopoiesis with somatic mutations in about 10% of older participants, with this frequency rising with age. In this context, since 2004 it has been suggested by some authors to screen older donors, who are more likely to carry hidden malignant clones, though most of these donors do not clinically develop into a hematologic malignancy. Data for such an approach are largely missing. Donor cell leukemia is very rare and it is therefore extremely difficult to define preventive strategies for this condition.

We expanded the discussion as follows (line 227-232): “A thorough assessment of the donor's family medical history is important, especially concerning leukemia and other hematological malignancies. This can be helpful to identify potential genetic predispositions in the donor. Donor cell leukemia is an exceedingly rare phenomenon, posing significant challenges in establishing preventive strategies for this condition.”

  1. Please indicate the age of the donor at the time of HSCT.

Authors: This is an important information indeed. The donor was 61 years old at the time of HSCT. We added the age of the donor accordingly (line 101): “the patient underwent subsequent allogeneic hematopoietic stem cell transplantation (allo-SCT) from her HLA-identical 61 year old brother in 06/2017.

Reviewer 3 Report

Comments and Suggestions for Authors

Dear Authors,

Thank you for your contribution. I read your manuscript where you described the case of a 43-year-old woman who developed donor-cell leukemia (DCL) as complication after a sibling (HLA-identical brother) HSCT. I think it is a relevant topic since, even if it is a rare condition, accounting for 0.1% of relapses, it needs for advanced diagnostic techniques for identification and for early detection, better prevention and treatment strategies.

I found only a mistake that should be changed: the hemoglobin value (line 75).

Kind regards

Author Response

Reviewer 3:

Thank you for your contribution. I read your manuscript where you described the case of a 43-year-old woman who developed donor-cell leukemia (DCL) as complication after a sibling (HLA-identical brother) HSCT. I think it is a relevant topic since, even if it is a rare condition, accounting for 0.1% of relapses, it needs for advanced diagnostic techniques for identification and for early detection, better prevention and treatment strategies.

I found only a mistake that should be changed: the hemoglobin value (line 75).

Authors: We fell much honored to receive such a positive feedback to our manuscript and are very grateful to the reviewer for that.

We changed the hemoglobin value unit to g/dL.

Round 2

Reviewer 1 Report

Comments and Suggestions for Authors

The authors quite well responded.